# Comparison of storage and lignin accumulation characteristics between two types of snow pea

Xuerui Li[1], Jinxiang Wang[2], Yunhui Qu[1], Yiping Li[1], Yasmin Humaira[3], Sajjad Muhammad[3], Hongmei Pu[1], Lijuan Yu[1]*, Hong Li[1]*

1 Agro-Products Processing Research Institute, Yunnan Academy of Agricultural Sciences, Kunming, Yunnan, China, 2 School of Life Sciences, Datong University, Datong, Shanxi, China, 3 Department of Biosciences, COMSATS University Islamabad (CUI), Islamabad, Pakistan

☯ These authors contributed equally to this work.
* ylj@yaas.org.cn (LY); ynveg@163.com (HL)

**Data Availability Statement:** All relevant data are within the manuscript and its Supporting Information files.

## Abstract

Snow pea is a very important vegetable, and its postharvest storage characteristics vary with species. Few studies on the differences in its storage characteristics are available. In this study, postharvest changes in metabolic rate (respiration rate and water loss rate), membrane permeability (relative conductivity), nutrient contents (total sugar, amino acids, starch), lignin, cellulose, β-Glucosidase (β-GC) enzyme activity, texture properties, PG enzyme activity and their relationship were analyzed in large sweet broad peas and small snow peas. On the 8th day of storage, we found that the respiration rate and water loss rate were increased, total sugars and total amino acids were decreased significantly in these two legume vegetables, and that metabolic rate was slower with less nutrients consumed in large sweet broad peas than in small snow peas. Throughout the 8-day whole storage, the lignin and cellulose contents were always lower in large sweet broad peas than in small snow peas. With the increasing storage time, small snow peas were more susceptible to lignification and fibrosis, which was observed in their texture properties. The enzyme activities related to cellulose and pectin degradation (β-GC, PG) also showed the same trend during the storage. At the late stage of storage, the taste of large sweet broad peas was better than that of small snow peas. In conclusion, the storage period of large sweet broad peas was longer than that of the small snow peas, and its lignification degree was lower than that of the small snow peas. Meanwhile, senescence and lignin accumulation led to hardening of snow pea during postharvest storage. Our findings provide a theoretical reference for improving the postharvest storage quality of snow pea and extending the shelf life.

## Introduction

Snow pea (*Pisum sativum* L. var. macrocarpon Ser.) is grown worldwide, extensively in northern USA, Europe, Canada, Russia, and China [1, 2]. Due to the delicious flavor and abundant nutrients, its tender shoot, seed, and pod are consumed as vegetables, especially the tender pod

**Funding:** The present research got supportion from the Yunnan Li Puwang Expert Workstation (202005AF50007). Yu Lijuan and Li Xuerui are the main accomplisher of this project, they design the study, collect and analyse dates, prepare the manuscripts; it also got supportion from Cultivation of Agricultural Product Processing Team (202002AE320007-03) and Green Food Brand Construction (Intensive Processing). Li Hong is the project leader, he designs this study. There was no additional external funding received for this study.

**Competing interests:** The authors have declared that no competing interests exist.

[3]. Its pod is one of the most important and popular leguminous vegetable because of high phytonutrients including amino acids, insoluble fibers, and vitamins [4]. However, its pod is highly perishable and ripened within a few days with a relatively short shelf-life, thus restricting its market potential.

Due to the high water content, strong metabolism, and microorganism reproduction, it is easy for pods to lose water, wrinkle, and then resulting in nutrient loss even deterioration during storage [5, 6]. Kader *et al.* have found that precooling and storing at near 0°C could extend the shelf life of pea pod. Kumar *et al.* have reported that cumin essential oil is preservative to extend shelf life of *Pisum sativum* [7]. Nasef *et al.* have found that microperforated polypropylene bags with 12 microholes are suitable to preserve pea pod, thus extending its retail period [8]. However, available information has been reported on the optimization of pea storage conditions and the information on the deterioration mechanism in pea is relatively scarce. El-hamahmy *et al.* have suggested that decay, shriveling, and reduction in vitamin C, and sugar contents are the primary factors limiting storage quality of snow pea pods [9].

Lignification is common in plants, and lignin can guarantee the integrity of cell wall, enhance the hardness and toughness of stem, provide plants with the internal mechanical support so that plants can resist abiotic and biotic stress [10, 11]. The lignification not only affects the taste and quality, but also limits the storage and transport of postharvest horticultural products. For example, lignin content exhibits an extremely significant negative correlation with edible quality [12]. The pea pods become hard and coarse with more fiber, which affects the taste of legumes.

In the present study, the changes in metabolic rate and nutrient contents were investigated to reveal deterioration mechanism in snow peas after harvest. The metabolic rate indexes significantly increased including the respiration rate, water loss rate, and electrical conductivity. The nutrient contents indexes rapidly declined including total sugar, total amino acids, and starch. Meanwhile, lignin and cellulose contents were ascended. Further, the texture properties such as hardness and chewiness changed. The aim of this study was to evaluate the storage characteristics of legume vegetables and lay the foundation for further research on related lignification and preservation technologies.

## Materials and methods

### Material treatments

Two types of snow pea including large sweet broad pea and small snow pea were purchased from the vegetable supermarket on the agricultural product display platform of Agro-product Processing Research Institute, Yunnan Academy of Agricultural Sciences (Kunming, China). Freshly picked pods were selected, and they were intact, free from pests and mechanical damage with similar size, maturity degree, and plumpness. Ten samples (300g per sample) with similar maturity were selected, and one group was the observation group, and the other group was the experimental group. The samples were collected on d 0, d 2, d 4, d 6, and d 8 during storage. All samples were placed at room temperature without direct sunlight, strong wind, and suitable humidity, and the samples was spread flat to keep ventilated. All of the treatments were conducted with three biological replicates.

### Assessments

To investigate the physiological and biochemical changes during storage, pods stored after 2, 4, 6, and 8 days were used as test samples. To avoid the possible error, the head and tail of selected pods were removed, and the middle part of pods was used for index measurement.

**Respiration rate.** Respiration rate was measured with SY-1022 Fruit and Vegetable Respiro meter. First, samples were put into the respiratory chamber for 30 min, the respiratory intensity of the sample was got according to the change of $CO_2$ concentration in the respiratory chamber before and after sample placement.

**Water loss rate.** The water loss rate was determined according to the change in the sample weight after harvest and during storage [13]. The initial weight of the sample was recorded as $M_0$, the weight of each sample at sampling time points was recorded as M. The water loss rate was computed according to the following formula:

$$Water\ loss\ rate\ (\%) = (M0 - M)/M0.$$

**Relative conductivity.** The samples were cut into small-sized pieces, and soaked in 20 mL of distilled water for 1 h. Afterwards, the initial electronic conductivity ($C_0$) was recorded, and then the samples were boiled for 30 minutes and cooled to record the electronic conductivity value ($C_1$). Finally, the electronic conductivity was computed as follows:

$$Relative\ conductivity\ (\%) = (C0/C1) * 100\%$$

**Total sugar content.** First, 2 g sample was added into 20 mL of distilled water, and then was boiled for 30min. The standard curve of glucose was constructed, y = 0.0037x+0.0928. Then it was determined according to the modified anthrone colorimetric method [8]. The absorbance value at 630 nm was measured. The total sugar content in the sample was calculated by using the following formula:

$$Total\ sugar\ content\ (\%) = (C * V1 * D)/ (W * V2 * 106) * 100\%$$

Whereas, C = sugar concentration (μg); $V_1$ = total volume of sugar extract (mL); D = dilution factor; W = total weight of sample (g); $V_2$ = volume of sugar used for the determination (mL).

**Total amino acid content.** The amino acid content was determined according to the improved ninhydrin colorimetric method [14]. A total of 5 g sample was added into 50 mL of distilled water and 5g activated carbon, and then was boiled, and filtered. Afterwards, 2 mL filtrate was added into distilled water (2 mL), 2% ninhydrin solution (1 mL), and phosphate buffer (1 mL), and boiled for 15 min. After cooling down, absorbance at 570 nm was measured.

**Starch content.** The starch content was determined with the plant starch content kit (Suzhou Keming Biotechnology Co., Ltd.). After measurement of the absorbance at 620 nm, starch content was calculated according to the following formula:

$$Starch\ content\ (mg/g\ fresh\ weight) = [(A + 0.0295) * V1]/2.936/(W * V1/V2)$$

Where A is the absorbance value; $V_1$ is the volume of the sample added to the reaction system; $V_2$ is the volume of extract added; W is the sample weight.

**Lignin content.** The sulfuric acid method was used to determine the lignin content [15]. The initial weight of the sample was recorded as $M_0$. The initial sample was added into 4 mL sulfuric acid (72%) and 200 mL distilled water, refluxed at 100˚C for 1 h, and filtered. The filter residue was washed with hot distilled water, and dried completely to a constant weight (M). The lignin content was calculated as follows:

$$Lignin\ content\ (\%) = (M/M0) * 100.$$

**Cellulose content.** Cellulose content was determined by the method of acidity washing.

The samples were added into 20 mL of cetyl trimethyl ammonium bromide. The remaining procedures were similar to those for the determination of lignin.

**Determination of enzyme activities of β-GC (β-Glucosidase) and PG (ploygalacturonase).** The 0.1 g of fresh snow pea tissue sample was added the extract and ground it in an ice bath. After high-speed centrifugation, the supernatant was taken and subjected to β-GC and PG enzyme activity determination with assay kits (Suzhou Keming Biotechnology Co., Ltd.), respectively, according to kit instruction. The absorbance value at 400 nm was measured, and 1 nmol p-nitrophenol produced per gram tissue per minute was defined as an enzyme activity unit of β-GC. The obtained supernatant was further diluted 5 times for PG determination, and the absorbance values were measured at a wavelength of 540 nm. One mg of galacturonic acid generated from pectic acid from per gram sample decomposition per hour at 40˚C and pH 6.0 was defined as an enzyme activity unit of PG (U).

**Texture properties.** The hardness, cohesiveness, springiness, gumminess, and chewiness were measured by the TMS-TOUCH texture analyzer equipped with a cylindrical detection probe (35 mm diameter). The test parameters were set as follows: the load unit of 200 N (Newton), the target deformation of 50%, the test speed of 60 mm/s, and the rising height (from the sample) of 10 mm, the initial force of 0.1 N, and the residence 0 s between two extrusion cycles. The measurement was repeated 10 times.

## Statistical analysis

All data were presented as the mean ± standard error (SE). The statistical difference between groups was analyzed through one-way analysis of variance (ANOVA) in SPSS software. $P<0.05$ was considered as statistically significant.

## Results and discussion

### Respiration rate and water loss

Water is essential to ensure the freshness of fruits and vegetables. When water loss exceeds 5%, fruits and vegetables usually begin to wilt and lose the freshness [16]. However, due to the strong respiration and transpiration, legumes are prone to lose water and wilt, and deteriorate. Hence, the respiration rate and water loss rate were examined in the present study (Fig 1). As shown in Fig 1A, the respiration rate was dramatically increased by 55.38% (from 66.54 to 103.39 mg·kg$^{-1}$·h$^{-1}$) for large sweet broad pea and by 139.75% (from 30.77 to 73.77 mg·kg$^{-1}$·h$^{-1}$) for small snow peas after 8-day storage, respectively. This indicated that the respiration change rate of the small snow peas was stronger. In addition, it was observed that the water loss rate of large sweet broad peas increased from 13.79% to 32.56%, and that of small snow peas increased from 15.82% to 36.03% (Fig 1B). The water loss rate of both types of peas increased during storage, while compared to large sweet broad peas, small snow peas is more likely to lose water. Above results indicated that respiration intensity of legumes increased with the extension of storage, which might be closely related with water loss.

### Relative conductivity

Relative conductivity is an important indicator for estimating the degree of destruction of cell membrane structure in fruits and vegetables in the course of storage [17]. Therefore, the current study analyzed the changes in relative conductivity of two snow pea during storage. As shown in Fig 2, the relative conductivity was increased by 30.68% (from initial 37.68% to 49.24%) for large sweet broad peas and by 47.33% (from 34.73% to 51.17%) for small snow

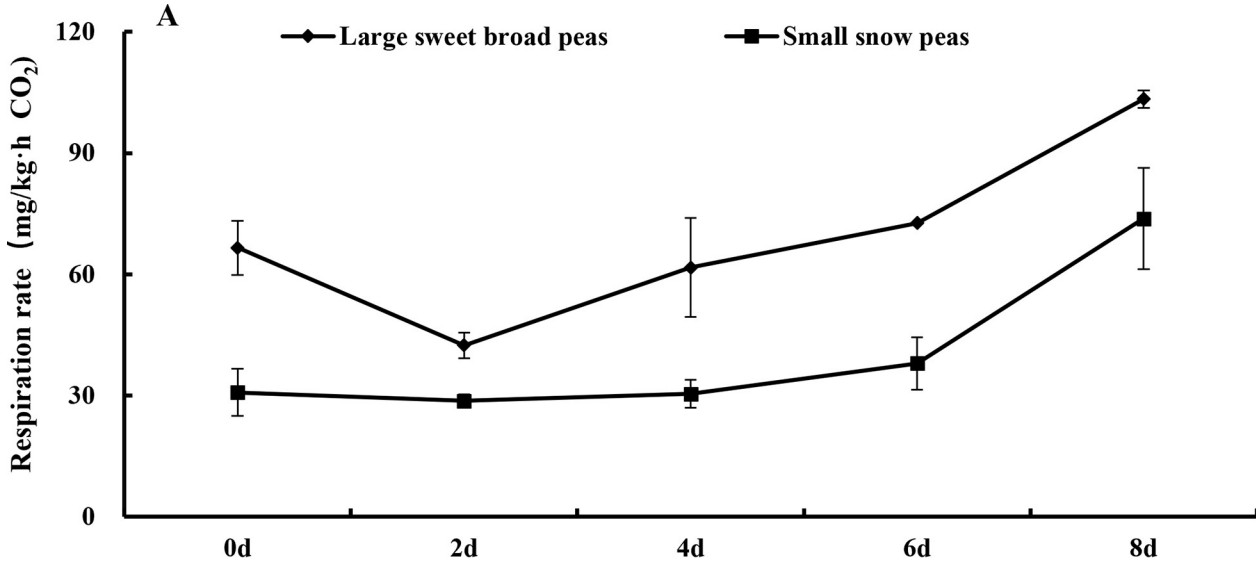

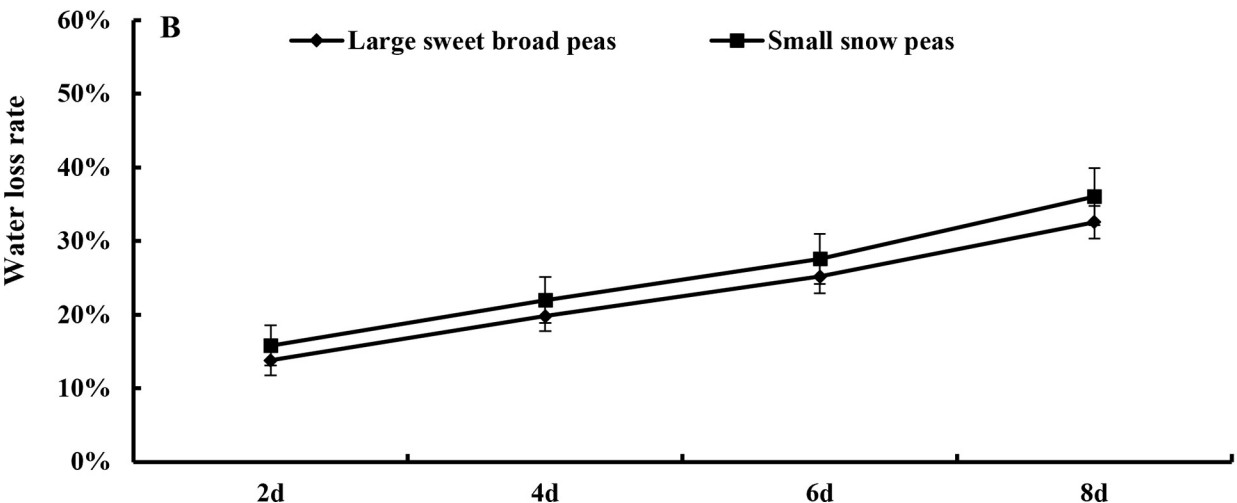

**Fig 1.** Changes in respiration rate (A) and water loss rate (B) of two types of snow peas during storage. Different lower-case letters indicate significant differences in mean among different treatment groups(n = 3)(P<0.05). The following was the same.

peas after storing for 8 days. This indicated that the cell membrane structure small snow peas is more drastically destructed than large sweet broad peas during postharvest storage.

## Total sugar, total amino acid, and total starch content

Sugar, amino acids, and starch are the main nutrients in legume vegetables. However, these nutrients were usually declined with the extension of storage [18]. Hence, in the current study, total sugar, total amino acids, and starch were investigated (Fig 3). As shown from Fig 3A and 3B, both the total sugar and total amino acid decreased with the extension of storage. The total sugar content declined relatively smooth from 4.42% to 3.35% in large sweet broad peas, whereas that in small snow pea declined sharply from 3.96% to 1.10% (Fig 3A). Thus, after 8-day storage, total sugar content of small snow pea was significantly lower than that of large sweet broad peas.

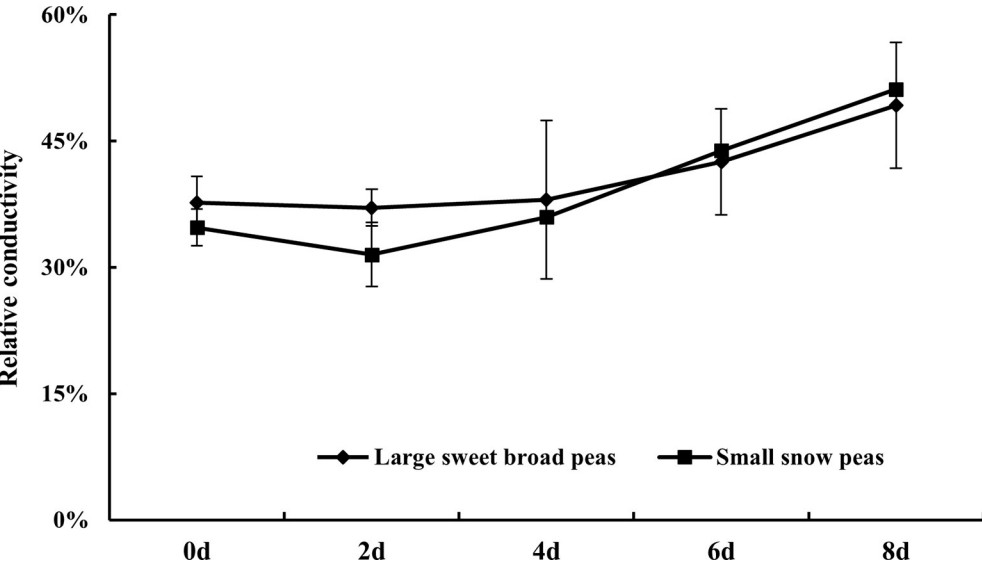

**Fig 2. Changes in relative conductivity of two types of snow peas during storage.**

As shown in Fig 3B, the change in amino acid content was consistent with that in total sugar content, which decreased respectively by 7.5% in large sweet broad peas and 14.1% in small snow peas. Although, there was a rise from 19.72 mg·g$^{-1}$ FW at the day 0 of storage to 29.14 mg·g$^{-1}$ FW at day 2 for large sweet broad pea, which might be caused by the late-maturing. The total amino acid content from 29.14 mg·g$^{-1}$ FW at the 2th day to 18.24 mg·g$^{-1}$ FW during the storage in large sweet broad peas. Comparatively, the total amino acid content declined smoothly in small snow peas, which decreased from 28.78 mg·g$^{-1}$ FW to 24.72 mg·g$^{-1}$ FW.

Also, the change in starch content was analyzed (Fig 3C). The starch content of large sweet broad peas increased from initial 14.19 mg·g$^{-1}$ FW to 17.45 mg·g$^{-1}$ FW on day 8, which might be attributed by after-ripening. While, the starch content of small snow peas decreased by 7.06 mg·g$^{-1}$ FW from 18.47 mg·g$^{-1}$ FW after harvest to 11.41mg·g$^{-1}$ FW on the 8th day of storage. According to above results, nourishments including the total sugar and the total amino acid was depleted in both two varieties after harvest. And it was indicated that the nutrients of small snow peas were consumed more quickly than large sweet broad peas.

## Lignin and cellulose content

Lignin and cellulose, as important components of plant secondary cell wall, participate in various physiological processes, excepically in the morphological development of plants [19]. β-PC

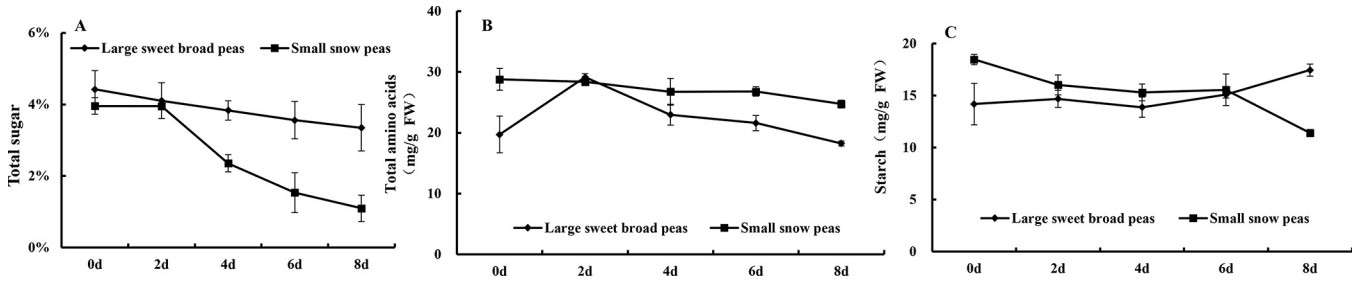

**Fig 3.** Changes in total sugar (A), total amino acid (B), and total starch (C) contents in two types of snow pea during storage.

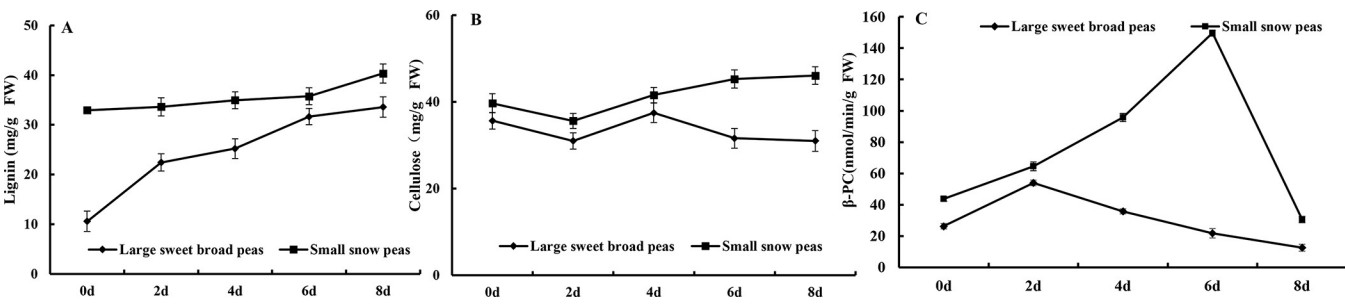

**Fig 4.** Changes in lignin content (A), cellulose content (B) and β-GC enzyme activity (C) of two types of snow peas during storage.

is the key enzymes responsible for cellulolysis, and inhibits the activity of enzymes related to the degradation of cell wall substances, thus delaying the decline in fruit hardness [20]. Lignification can ensure the integrity of cell wall, enhance the hardness and toughness of stem, and provide plants with the internal mechanical support so that plants can resist abiotic and biotic stresses [10]. However, the accumulation of lignin and cellulose not only affects the taste and quality of fruits and vegetables, but also limits their postharvest storage and transportation. It has been reported that fruits and vegetables become hardened due to the increased lignin and cellulose during storage or under stress conditions [21].

In the present study, lignin content in both types of peas significantly increased with the extension of the storage (Fig 4A). Although, the lignin content in large sweet broad peas raised from 10.60 to 33.57 mg·g$^{-1}$ after 8-day storage, but the lignin content in large sweet broad peas was always lower than that in small snow peas. This indicated that the small snow peas were more easily lignified with the increasing storage. The cellulose between two types of peas exhibited the similar regularity as lignin content (Fig 4B). In large sweet broad peas, the cellulose content increased from 35.65 mg·g$^{-1}$ to 37.48 mg·g$^{-1}$ in the first 4 days of storage, and then decreased to 31.01 mg·g$^{-1}$ in the last 4 days of storage, which might be due to the degradation of protopectin in the postharvest. However, the cellulose content increased from 39.67 mg·g$^{-1}$ to 46.06 mg·g$^{-1}$ after storage in the small snow peas during 8-day storage. In the processing of storage, the cellulose content in large sweet broad peas was also always lower than that in small snow peas.

As shown in Fig 4C, the enzyme activity of β-PC enzyme related to cellulose degradation, of large sweet broad peas reached the highest (53.98 nmol/min/g) on day 2 during storage, and then decreased continuously, and reduced by 51.89% during the entire storage period. The β-PC enzyme activity of small sweet broad peas increased continuously until 149.61 nmol/min/g in the first 6 days, and then decreased to 30.55 nmol/min/g. During the whole storage process, the β-PC enzyme activity of small sweet broad peas was greater than that of large sweet broad peas, and the change trend of β-PC enzyme activity was consistent with that of the cellulose content. Small sweet broad peas were more prone to lignification during storage and were not resistant to storage.

## Texture characteristics and PG enzyme activity

With the increase in lignin and cellulose contents during the storage, the textural properties of fruits and vegetables changed greatly. The textural properties of fruits and vegetables are also important for evaluating their quality and economic values [22]. Usually, the texture profile analysis (TPA) method was used to simulate the chewing motion of human teeth to analyse the textural properties, in which the sample was compressed twice to test its cohesiveness,

springiness, gumminess, and chewiness. This texture evaluation method reduced the subjective evaluation error to a certain extent [23].

In this study, the TPA method was used to analyze changes in textural properties of the two types of peas during storage. The results indicated that the cohesiveness of two types of peas was increased by 21.83% from 0.20 to 0.24 for large sweet broad peas and increased by 121.63% for small snow peas from 0.17 to 0.39. This corresponded exactly to the water loss rate, and a large amount of water loss caused the internal cohesiveness to increase.

The springiness of the two types of peas showed an upward trend until the 6th day of storage, increasing from 3.023 to 4.06 mm for large sweet broad peas, and from 2.605 to 3.7 mm for small snow peas, and then springiness declined relatively smoothly in both types of peas. Overall, in 8-day storage, springiness increased by 8.9% for large sweet broad peas and by 33.05% for small snow peas.

The gumminess and chewiness of the two types of peas showed a rapid upward trend during 8-day storage. In large sweet broad peas, the gumminess increased by 57.53% from 1.00 to 2.73 N, and the chewiness increased by 209.13% from 3.021 to 9.30 N. On the 8th day, gumminess and chewiness values of large sweet broad peas were slightly decreased, which was similar to springiness in large sweet broad peas, and this decrease might be attributed to the consumption of nutrients in the late storage stage. In small snow peas, the gumminess increased by 245.90% from 1.07 to 3.70 N, and the chewiness increased by 375.53% from 2.73 to 12.97 N. Compare with the large sweet broad peas, the rising trend of chewiness in small snow peas was more obvious.

PG is related to the degradation of pectin during fruit softening. Koutsimanis et al. have showed that 1 μL/L 1-MCP treatment could effectively inhibit the decrease in PG activity and protopectin content of Tardibelle peach fruit during shelf life at 20˚C, and maintain the fruit higher hardness [24]. As shown in Fig 5C, the PG enzyme activity of large sweet broad peas increased from 7.04 to 31.74 mg/h/g on day 4 during storage, which was consistent with the increase in their cellulose content. On day 8, PG enzyme activity of large sweet broad peas decreased to 9.8 mg/h/g. During the whole storage process, their PG enzyme activity increased by 39.17%. The PG enzyme activity of small sweet broad peas reached the highest on day 6 (9.27mg/h/g), and then decreased slightly, and it increased by 101.30% during the whole storage process. This result indicated that the small sweet broad peas were easier to soften during storage and were not resistant to storage, which was consistent with the analysis results of texture.

## Conclusion

This study revealed that small snow peas had a fast metabolism and consumed more nutrients than large sweet broad peas after harvest, resulting in increased membrane permeability and

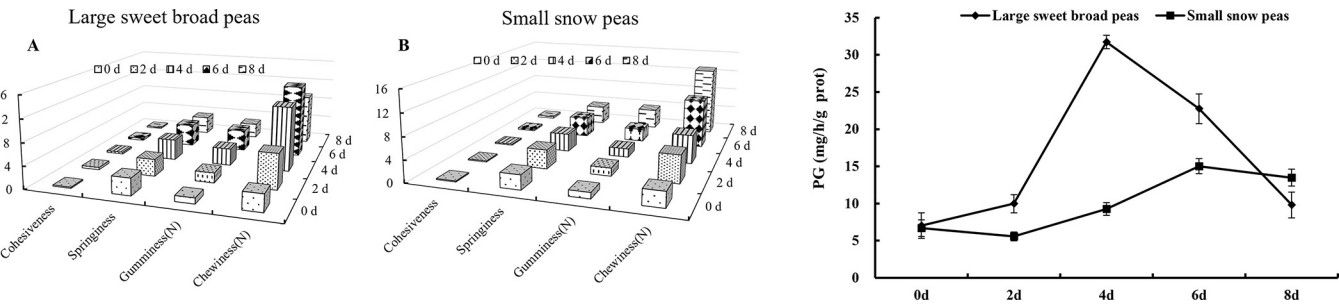

**Fig 5.** Changes in texture characteristics AB) and PG enzyme activity (C) of legume vegetables during storage.

greater perishability. In addition, postharvest lignification degree of small sweet broad peas was higher than that of large ones, and their cellulose content and related enzyme activities were higher than those of large sweet broad peas, which affected their flavor and shelf life. The texture properties and PG enzyme activity related to pectin degradation also verified this. Overall, the large sweet broad peas had a longer shelf life than the small ones, and their lignification degree was lower than that of the small ones. Future research is suggested to focus on the molecular biological mechanism of pod lignification during storage and to explore preservation methods to alleviate pod lignification. Our findings provide a theoretical basis for the preservation of legume vegetables.

## Supporting information

**S1 File. The underlying date.**
(XLSX)

## Author Contributions

**Conceptualization:** Lijuan Yu, Hong Li.

**Data curation:** Xuerui Li, Yiping Li, Lijuan Yu.

**Formal analysis:** Xuerui Li, Sajjad Muhammad, Lijuan Yu.

**Funding acquisition:** Yunhui Qu, Hong Li.

**Investigation:** Jinxiang Wang.

**Resources:** Yasmin Humaira, Hongmei Pu.

**Supervision:** Hong Li.

**Writing – original draft:** Xuerui Li.

**Writing – review & editing:** Xuerui Li, Lijuan Yu.

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
