## [Decision Letter · Decision Letter 0]

17 Feb 2022

PONE-D-21-40678Senescence and lignin accumulation lead to hardening of snow pea during postharvest storagePLOS ONE

Dear Dr. Li,

Thank you for submitting your manuscript to PLOS ONE. After careful consideration, we feel that it has merit but does not fully meet PLOS ONE’s publication criteria as it currently stands. Therefore, we invite you to submit a revised version of the manuscript that addresses the points raised during the review process. Please submit your revised manuscript by Apr 03 2022 11:59PM. If you will need more time than this to complete your revisions, please reply to this message or contact the journal office at plosone@plos.org. Please include the following items when submitting your revised manuscript:A rebuttal letter that responds to each point raised by the academic editor and reviewer(s). You should upload this letter as a separate file labeled 'Response to Reviewers'.A marked-up copy of your manuscript that highlights changes made to the original version. You should upload this as a separate file labeled 'Revised Manuscript with Track Changes'.An unmarked version of your revised paper without tracked changes. You should upload this as a separate file labeled 'Manuscript'.

We look forward to receiving your revised manuscript.

Kind regards,

Sajid Ali

Academic Editor

PLOS ONE

Journal Requirements:

(The present research got supportion from the Yunnan Li Puwang Expert Workstation (202005AF50007). Yu Lijuan and Li Xuerui are  the main accomplisher of this project,they design the study, collect and analyse dates,prepare the manuscripts;

it also got supportion from Cultivation of Agricultural Product Processing Team (202002AE320007-03).Li Hong is the project director, he design this study.)

(None of our co-authors has conflict of interest to declare.)

Additional Editor Comments:

Please revise your manuscript carefully by following the comments of the reviewers. The comments of the reviewer 1 are very critical and need to be addressed carefully. 

Reviewers' comments:

Reviewer's Responses to Questions

**Comments to the Author**

1. Is the manuscript technically sound, and do the data support the conclusions?

Reviewer #1: No

Reviewer #2: Yes

Reviewer #3: Yes

2. Has the statistical analysis been performed appropriately and rigorously? 

Reviewer #1: No

Reviewer #2: No

Reviewer #3: I Don't Know

3. Have the authors made all data underlying the findings in their manuscript fully available?

Reviewer #1: No

Reviewer #2: Yes

Reviewer #3: Yes

4. Is the manuscript presented in an intelligible fashion and written in standard English?

Reviewer #1: No

Reviewer #2: No

Reviewer #3: Yes

5. Review Comments to the Author

Reviewer #1: In the manuscript PONE-D-21-40678, the authors investigated the senescence and lignin accumulation lead to hardening of snow pea during postharvest storage.

Overall, the methodology used by the authors about two varieties of snow pea and the targeted measurements are rather basic, and hence do not provide any additional information to what is already known about the biochemical changes in pea. There are a number of issues of concern and limited amount of new science in the manuscript. Authors are advised to find out more metabolites for comparison of the two varieties of snow pea, as the quality related measurements are very basic. Cell wall degrading enzymes, browning indexes and cell damage indexes would support more precisely the story.

Taking all the factors into account, I believe the manuscript is not suitable for publication in its present state in Journal of Plos One.

Some specific comments are given below:

Title: The title not complete reflects the content of the manuscript.

Line No. 66-75: In the present study, the changes in metabolic rate and nutrient contents were investigated to reveal deterioration mechanism in super snap peas after harvest. The targeted pea is not super snap peas in this study. The aim of this study is very general. In the introduction, no needs to write the results of this study.

Line No. 89-90: First, samples were respectively put into the respiratory chamber, after standing for 0.5 h. What did the authors mean by 0.5h?

Line No. 145-148: Pease, write the experimental design used in this study.

Line No. 155-159: As shown in Fig 1A, the respiration rate was dramatically increased by 55.38% (from 66.54 to 103.39 mg•kg-1•h-1) for large sweet broad pea and by 139.75% (from 30.77 to 73.77 mg•kg-1•h-1) for small snow peas after 8-day storage, respectively. This indicated that the respiration rate of the small snow peas was stronger. There is contradiction in you results and Fig. 1A. The Fig. 1A, indicates that the respiration rate was higher in Large sweet broad peas as compared to Small snow peas. However, the authors claimed in the results that the respiration rate was higher in Small snow peas. If the respiration rate was higher in the Large sweet broad peas as indicted in the Fig. 1A, then how the authors concluded that the shelf life of Large sweet broad was longer compared to Small snow peas?

The authors stated simple concepts and common knowledge over and over again. The results should be properly discussed in view of literature data.

Fig. 1, 2 and 3 should be constructed like Fig. 4.

Please, improve the conclusion by adding some salient and new findings of the present study.

Reviewer #2: The manuscript no PONE-D-21-40678 entitled 'Senescence and lignin accumulation lead to hardening of snow pea during postharvest storage' addressed nutritional, organoleptic and lignification changes in two varieties of snow pea after harvest. The study was very interesting and of economic importance for stakeholders, retailers, processors and exporters of pea industry. However, the manuscript needs some minor revisions in methodology section and overall the manuscript should be checked for grammatically mistakes and language improvement. I have suggested some changes in the manuscript pasted in 'Reviewer attachment'. After incorporation of these changes the manuscript can be considered for publication.

Reviewer #3: Research presented in the manuscript investigates changes in respiration, water loss, membrane permeability, nutrient contents, lignin contents, cellulose contents, and textural attributes in two snow pea varieties during storage. Findings suggest that lignin accumulation leads to hardening of snow pea during postharvest storage. Though, data presented provide adequate evidence to support the claim, however, few basic elements of research are missing and must be included in the manuscript. Storage conditions, replication size (sample weight), number of replications allocated to each removal and experimental design are missing in the "Materials and Methods" section. Figure captions must also state description of standard error bars used in graphs? Overall, manuscript is written well and needs only few corrections highlighted in the reviewed version of manuscript.

6. PLOS authors have the option to publish the peer review history of their article (what does this mean?). If published, this will include your full peer review and any attached files.

Reviewer #1: **Yes: **Ghulam Khaliq

Reviewer #2: No

Reviewer #3: No

---

## [Author Response · Author response to Decision Letter 0]

6 Apr 2022

-RESPONSE To REVIEWERS

> Dear Sajid Ali,

> Thank you very much for your kind consideration. Thanks a lot for the reviewers’ comments and their kind suggestions on our manuscript (PONE-D-21-40678). We provide this cover letter to explain, point by point, the details of our revisions in the manuscript and our responses to the reviewers’ comments as follows. In order to make the changes easily viewable for you and the reviewers, in the revised paper, we marked the revision with red color. Besides, we have carefully proof-read the manuscript to minimize typographical, grammatical and bibliographical errors. 

> We are looking forward to hearing from you soon.

> Best regards,

> Xuerui Li

Journal Requirements:

We have revised my manuscript meets PLOS ONE's style requirements, including those for file naming. You can see them in the chapter of References.

(The present research got supportion from the Yunnan Li Puwang Expert Workstation (202005AF50007). Yu Lijuan and Li Xuerui are the main accomplisher of this project, they design the study, collect and analyse dates,prepare the manuscripts; it also got supportion from Cultivation of Agricultural Product Processing Team (202002AE320007-03). Li Hong is the project director, he designed this study.)

The present research got supportion from the Yunnan Li Puwang Expert Workstation (202005AF50007). Yu Lijuan and Li Xuerui are the main accomplisher of this project, they design the study, collect and analyse dates, prepare the manuscripts; it also got supportion from Cultivation of Agricultural Product Processing Team (202002AE320007-03) and Green Food Brand Construction (Intensive Processing). Li Hong is the project leader, he designs this study. There was no additional external funding received for this study.

We have added the funding “Green Food Brand Construction (Intensive Processing)” and statement “There was no additional external funding received for this study.” in my updated Funding Statement. 

(None of our co-authors has conflict of interest to declare.)

We have stated “The authors have declared that no competing interests exist.” in our cover letter and mark the revision with red color.

We have established a supporting information file about our study’s minimal underlying data.

 We have added the caption for our supporting information files at the end of our manuscript.

Additional Editor Comments:

Please revise your manuscript carefully by following the comments of the reviewers. The comments of the reviewer 1 are very critical and need to be addressed carefully. 

Comments to the Author

5. Review Comments to the Author

Reviewer #1: In the manuscript PONE-D-21-40678, the authors investigated the senescence and lignin accumulation lead to hardening of snow pea during postharvest storage.

Overall, the methodology used by the authors about two varieties of snow pea and the targeted measurements are rather basic, and hence do not provide any additional information to what is already known about the biochemical changes in pea. There are a number of issues of concern and limited amount of new science in the manuscript. Authors are advised to find out more metabolites for comparison of the two varieties of snow pea, as the quality related measurements are very basic. Cell wall degrading enzymes, browning indexes and cell damage indexes would support more precisely the story.

 Thank you for your excellent suggestion and supporting. We have added the dates and discusses about cell wall degrading enzyme activities (β-GC, PG) in Fig 4.C and Fig 5.C.

Some specific comments are given below:

Title: The title not complete reflects the content of the manuscript.

Thank you for your excellent suggestion. The title was revised to “Comparison of storage and lignin accumulation characteristics between two types of snow pea”.

Line No. 66-75: In the present study, the changes in metabolic rate and nutrient contents were investigated to reveal deterioration mechanism in super snap peas after harvest. The targeted pea is not super snap peas in this study. The aim of this study is very general. In the introduction, no needs to write the results of this study.

In the present study, the changes in metabolic rate and nutrient contents were investigated to reveal deterioration mechanism in snap peas after harvest.

We have deleted “super”. The aim of this study is Analysis of storage characteristics and lignin accumulation of two snow pea (Large sweet broad peas and small snow peas).

Meanwhile, we have deleted the sentence “These results imply that lignification in snow pea may cause the change in texture properties, thus affecting its taste and quality” in the introduction.

Line No. 89-90: First, samples were respectively put into the respiratory chamber, after standing for 0.5 h. What did the authors mean by 0.5h?

It means 30 min.

Line No. 145-148: Pease, write the experimental design used in this study.

Thank you for your suggestion, we have revised it in the chapter 2 (Materials and methods).

Material treatments

Two types of snow pea including large sweet broad pea and small snow pea were purchased from the vegetable supermarket on the agricultural product display platform of Agro-product Processing Research Institute, Yunnan Academy of Agricultural Sciences (Kunming, China). Freshly picked pods were selected, and they were intact, free from pests and mechanical damage with similar size, maturity degree, and plumpness. Ten samples (300g per sample) with similar maturity were selected, and one group was the observation group, and the other group was the experimental group. The samples were collected on d 0, d 2, d 4, d 6, and d 8 during storage. All samples were placed at room temperature without direct sunlight, strong wind, and suitable humidity, and the samples was spread flat to keep ventilated. All of the treatments were conducted with three biological replicates.

Line No. 155-159: As shown in Fig 1A, the respiration rate was dramatically increased by 55.38% (from 66.54 to 103.39 mg•kg-1•h-1) for large sweet broad pea and by 139.75% (from 30.77 to 73.77 mg•kg-1•h-1) for small snow peas after 8-day storage, respectively. This indicated that the respiration rate of the small snow peas was stronger. There is contradiction in you results and Fig. 1A. The Fig. 1A, indicates that the respiration rate was higher in Large sweet broad peas as compared to Small snow peas. However, the authors claimed in the results that the respiration rate was higher in Small snow peas. If the respiration rate was higher in the Large sweet broad peas as indicted in the Fig. 1A, then how the authors concluded that the shelf life of Large sweet broad was longer compared to Small snow peas?

The authors stated simple concepts and common knowledge over and over again. The results should be properly discussed in view of literature data.

Thank you for your suggestion. What we want to express was the rate of respiration change, which is the characterization index of postharvest aging. The faster the respiration rate changes, the more vigorous the metabolism, the more susceptible to decay. 

So, we revised the sentence“This indicated that the respiration change rate of the small snow peas was stronger. ”

There are no studies comparing the storage characteristics of these two snow peas.

Fig. 1, 2 and 3 should be constructed like Fig. 4.

Thank you for your suggestion. We have revised them. The texture properties are shown more tellingly in Fig.5 AB.

Please, improve the conclusion by adding some salient and new findings of the present study.

Thank you for your suggestion. Some Chinese researchers have looked the storage characteristics of legumes, but there are no studies comparing the storage characteristics of these two snow peas. We found that the large sweet broad peas had a longer shelf life and a lower lignification degree than small snow peas. The enzyme activities related to cellulose and pectin degradation (β-GC, PG) also showed the same trend during the storage.

Reviewer #2: The manuscript no PONE-D-21-40678 entitled 'Senescence and lignin accumulation lead to hardening of snow pea during postharvest storage' addressed nutritional, organoleptic and lignification changes in two varieties of snow pea after harvest. The study was very interesting and of economic importance for stakeholders, retailers, processors and exporters of pea industry. However, the manuscript needs some minor revisions in methodology section and overall the manuscript should be checked for grammatically mistakes and language improvement. I have suggested some changes in the manuscript pasted in 'Reviewer attachment'. After incorporation of these changes the manuscript can be considered for publication.

Thank you for your excellent suggestion and supporting. We have revised grammatically mistakes according to your suggestion in 'Reviewer attachment'. Meanwhile, language was edited and polished by linguistics professor Ping Liu from Huazhong Agriculture University, Wuhan, China. 

Reviewer #3: Research presented in the manuscript investigates changes in respiration, water loss, membrane permeability, nutrient contents, lignin contents, cellulose contents, and textural attributes in two snow pea varieties during storage. Findings suggest that lignin accumulation leads to hardening of snow pea during postharvest storage. Though, data presented provide adequate evidence to support the claim, however, few basic elements of research are missing and must be included in the manuscript. Storage conditions, replication size (sample weight), number of replications allocated to each removal and experimental design are missing in the "Materials and Methods" section. Figure captions must also state description of standard error bars used in graphs? Overall, manuscript is written well and needs only few corrections highlighted in the reviewed version of manuscript.

Thank you for your excellent suggestion and supporting. we have revised it in the chapter 2 (Materials and methods). It concluded storage conditions, replication size (sample weight), number of replications allocated to each removal and experimental design.

Material treatments

Two types of snow pea including large sweet broad pea and small snow pea were purchased from the vegetable supermarket on the agricultural product display platform of Agro-product Processing Research Institute, Yunnan Academy of Agricultural Sciences (Kunming, China). Freshly picked pods were selected, and they were intact, free from pests and mechanical damage with similar size, maturity degree, and plumpness. Ten samples (300g per sample) with similar maturity were selected, and one group was the observation group, and the other group was the experimental group. The samples were collected on d 0, d 2, d 4, d 6, and d 8 during storage. All samples were placed at room temperature without direct sunlight, strong wind, and suitable humidity, and the samples was spread flat to keep ventilated. All of the treatments were conducted with three biological replicates.

 Figure captions have also stated description of standard error bars used in graphs. 

Different lowercase letters indicate significant differences in mean among different treatment groups(n=3) (P<0.05), The following was the same.

---

## [Editor Report · Decision Letter 1]

9 May 2022

Comparison of storage and lignin accumulation characteristics between two types of snow pea

PONE-D-21-40678R1

Dear Dr. Li,

We’re pleased to inform you that your manuscript has been judged scientifically suitable for publication and will be formally accepted for publication once it meets all outstanding technical requirements.

Kind regards,

Sajid Ali

Academic Editor

PLOS ONE
---

## [Editor Report · Acceptance letter]

22 Jun 2022

PONE-D-21-40678R1 

Comparison of storage and lignin accumulation characteristics between two types of snow pea 

Dear Dr. Li:

I'm pleased to inform you that your manuscript has been deemed suitable for publication in PLOS ONE. Congratulations! Your manuscript is now with our production department. 

Kind regards, 

on behalf of

Dr. Sajid Ali 

Academic Editor

PLOS ONE